# Uncovering the Heterogeneity in Fitness App Use: A Latent Class Analysis of Chinese Users

**DOI:** 10.3390/ijerph191710679

**Published:** 2022-08-27

**Authors:** Li Crystal Jiang, Mengru Sun, Guanxiong Huang

**Affiliations:** 1Department of Media and Communication, City University of Hong Kong, Hong Kong SAR, China; 2College of Media and International Culture, Zhejiang University, Hangzhou 310058, China

**Keywords:** fitness app, physical activity, latent class analysis, WeRun, motivation, self-determination

## Abstract

This study examines fitness app use patterns and their correlates among Chinese users from the perspectives of uses and gratification theory and self-determination theory. Our sample comprised 632 users of WeRun, the fitness plugin of WeChat, the largest Chinese mobile social networking app; participants completed an online survey and provided self-tracked physical activity data, which were subjected to latent class analysis. Based on the four-class latent class model (which yielded the best model fit and the most interpretable results), 30.5%, 27.5%, 24.7%, and 17.3% of the users were categorized as light users, reward-oriented users, lifestyle-oriented users, and interaction-oriented users, respectively. Moreover, class membership was associated with gender, age, education, income, life satisfaction, autonomy, and platform-based motivations. There is a significant heterogeneity in fitness app use and exercise behaviors. Platform-based motivations and autonomy are important classification factors, as users are looking for specific kinds of gratification from their use of fitness apps. Demographics and individual characteristics are also explanatory factors for class membership. The study findings suggest that fitness app designers should segment users based on motivation and gratification.

## 1. Introduction

Since the beginning of the 21st century, physical activity levels have declined globally [1,2,3]. It is estimated that about one-third of adults around the world are not physically active, which can result in a number of health risks [1]. Yet mobile apps that focus on exercise and weight loss are also becoming increasingly popular, accounting for more than 70% of total app usage in the health and fitness category [4,5,6].

Previous research has shown that the use of mobile fitness apps can effectively promote exercise [7,8,9,10,11]. Previous studies have also documented significant heterogeneity in both fitness app usage and participation in physical activity. On the one hand, individual differences have been reported in fitness app usage, motivations, and gratifications. For example, there are some gender-specific differences in fitness app adoption and gratification [12]. Users also have different goal orientations [13] and function preferences [14], leading to diverse patterns of fitness app usage. On the other hand, differences in exercise behaviors have been noted based on gender, age, exercise-identity (for a review, see [15,16]), and other personal dispositions [17,18,19]. Individuals also report multiple motives for participating in physical activity, including fitness-related motives and social purposes [18]. However, relatively little is known about whether and how these heterogeneities in app usage patterns and exercise behaviors are correlated. Previous research is atheoretical and descriptive, which makes it difficult to promote changes among different behavioral patterns using psychosocial mechanisms. Moreover, previous research has mostly captured the heterogeneity through observed variables (e.g., exercise behaviors, demographics). It remains unclear whether we can use remediable factors to describe different behavioral patterns. Individual needs and motivations are recognized as one of the most remediable factors in shaping health behaviors; thus, it is possible to use them to understand the heterogeneities in app usage patterns and exercise behaviors.

These gaps in the extant research have motivated the present study, which seeks to develop a more nuanced understanding by examining behavioral heterogeneity in fitness app use and exercise behaviors through theoretical lenses. We aim to uncover the subgroups in user profiles and identify theoretical correlates of these group profiles through a person-centered approach, as has been widely used to capture population heterogeneity in health beliefs and behaviors [20,21,22]. Following previous profiling exercises, we first gathered data regarding fitness app uses and exercise behaviors to conduct a latent class analysis (LCA); this was used to identify meaningful subgroups. Then, we considered subgroup membership from the perspective of uses and gratification theory and self-determination theory, by predicting the membership with different sets of motives and demographic variables.

For this purpose, we employed WeRun, the fitness plugin embedded in WeChat, which had more than one billion active users in 2020, and is the most widely used mobile social networking app in China [23]. We chose WeRun as the app allows individuals to build their fitness community based on their WeChat networks. Besides the basic function of monitoring step counts, WeRun also allows for multiple social processes. Specifically, it supports social comparison by ranking WeRun users and their friends by step counts; it incorporates social support by allowing the users to like each other’s step counts; and it leverages gamification mechanisms by providing WeRun users the chance to monetize their step counts or donate their step counts to charity [24,25].

Our inquiry was guided by three research questions: (1) how do different WeRun user subgroups differ in their fitness app use and exercise behaviors? (2) What are the motivations that can predict subgroup membership among WeRun users? And (3) are WeRun user demographics predictive of subgroup membership? The answers to these questions will offer a comprehensive segmentation of fitness app users and elucidate the practical implications for the design of fitness apps and physical activity interventions.

Like other human behaviors, both mobile app use and exercise behaviors are motivated by specific needs. Several theoretical lenses can thus be applied to better understand the motivations behind fitness app usage. For example, according to the uses and gratification theory (U&G), which was originally developed to explain traditional media use [26], the usage of any medium is contingent on the extent to which it can meet a particular need. In the past decade, this theory has been extensively applied to understand the gratifications provided by social media use [27]. Findings yielded by these investigations indicate that psychological needs (e.g., entertainment, relaxation, information seeking) and social gratifications (e.g., interpersonal bonding, status seeking) are the primary motivations underlying engagement with social media [28,29]. In the context of mobile fitness apps, however, gratifications tend to be examined in light of their affordances [30,31,32]. Following this practice in the present study, based on the major functions of fitness apps, we classified the platform-based motivations into five categories, denoted as exercise, comparison, social, financial, and charity motivations.

We also draw upon self-determination theory, which posits that individuals make free choices about their own actions, based on a full understanding of their own needs and environmental information [33]. The theory further conceptualizes human needs in terms of intrinsic and extrinsic needs, which fundamentally differ in the strength of autonomy [34]. Specifically, individuals with high levels of autonomy perceive the inherent interest of and satisfaction that might be derived from an activity, whereas those with low autonomy are guided by the exterior rewards or punishments attached to the same activity.

Although originating from different research domains, both the uses and gratification theory and the self-determination theory purport that individuals are goal-oriented and self-determined. As a result, previous research has applied the two in conjunction to examine computer-mediated communication [35] and fitness app adoption [12]. In this study, given that the two theories provide unique insights into fitness app use and exercise behaviors, we integrate the two perspectives for a more sophisticated understanding of WeRun users’ needs and gratifications.

## 2. Materials and Methods

### 2.1. Design and Procedure

As the aim of the study was to establish WeRun use patterns and identify the correlates of these patterns, in January 2021, WeRun users were invited to take part in an online survey. The survey was administrated by one of the largest online survey firms in China, which had a subject pool of more than 2.6 million Chinese participants with a wide range of demographics. The eligibility criteria were set to WeRun users aged 18 and above. Special caution was taken in the recruitment to cover different age groups and both genders to maximize sample representativeness. The survey company sent the invitation to registered participants in the sampling pool randomly, in order to reach a broad range of age and gender groups. Prior to conducting the data collection, approval from the Institutional Review Board of the university with which the authors are affiliated was obtained.

### 2.2. Measures

Fitness app use behaviors: WeRun provides tracking, ranking, liking, and awarding functions to its users. The users can track their daily step counts and step rankings within their WeRun networks, like others’ step counts and receive likes from others, and earn awards by donating steps to charity or monetizing their step counts. Accordingly, we measured fitness app use with five items that captured common activities in WeRun, each requiring a response on a 5-point scale (1 = never, 5 = always). Specifically, the participants were asked to indicate the extent to which they checked step count rankings, liked others’ step counts, received likes from others, donated steps to charity, and monetized their step counts (M = 3.20, SD = 0.44, Cronbach’s α = 0.80).

Exercise behaviors: we also asked the participants to report how many times per week they performed different types of moderate- to high-intensity physical activity (e.g., brisk walking, water aerobics, hiking; M = 3.58, SD = 1.62) and how long each session lasted (M = 62.66 min, SD = 34.88). They were also instructed to check their WeRun records and report their daily step counts in the past week. We created an average score of daily step counts (M = 8446.05, SD = 7263.71) to facilitate data analysis.

Autonomy: as one of the key constructs in self-determination theory, autonomy was measured according to the four items (“highly compatible with my choices and interests,” “fits perfectly the way I prefer to exercise,” “definitely an expression of myself,” and “affording the opportunity to make choices”) adapted from a previous study [36]. Each item required a response on a 5-point scale, ranging from 1 = “totally disagree” to 5 = “totally agree” (M = 3.78, SD = 0.67, Cronbach’s α = 0.72).

Platform-based motivations: based on the categories proposed by previous literature [30], we measured five specific sets of motivations for using WeRun on a five-point scale (1 = totally disagree, 5 = totally agree). The categories included exercise motivations (two items; M = 3.90, SD = 0.74, Spearman–Brown coefficient = 0.56; e.g., “I use WeRun to push myself to exercise more”), social motivations (six items; M = 3.63, SD = 0.70, Cronbach’s α = 0.78; e.g., “I use WeRun to connect with my friends”), financial motivations (three items; M = 3.21, SD = 0.93, Cronbach’s α = 0.76; e.g., “I use WeRun to earn some money by trading my steps for cash”), charitable motivations (three items; M = 3.66, SD = 0.80, Cronbach’s α = 0.65; e.g., “I use WeRun so I can help others by donating my steps to charity”), and comparison motivations (five items; M = 4.38, SD = 0.88, Cronbach’s α = 0.68; e.g., “I use WeRun to compare my number of steps with my friends”).

Life satisfaction: participants were asked to rate on a 5-point scale (1 = totally disagree, 5 = totally agree) their satisfaction with life using five items (e.g., “I’m quite content with my life at the moment”) adapted from the instrument developed by Diener et al. [37] (M = 3.26, SD = 0.87, Cronbach’s α = 0.86).

In addition to the variables used in the present study, the survey also included items on body satisfaction, social network size, social comparison, and social support, as reported elsewhere [38,39].

### 2.3. Data Analysis

After dummy-coding the items measuring WeRun use and exercise behaviors, we conducted LCA to identify categorical classes [40]. The LCA method also allows class membership to be predicted based on using autonomy, platform-based motivations, life satisfaction, and demographics as predictors. LCA was performed using the poLCA package in R statistical software [41]. We estimated a series of LCA models with robust maximum likelihood methods, which maximize the between-class difference and within-class similarity. To reduce the chance of obtaining local maxima, when performing the iterative maximization procedure, we used different random sets of starting values. The model fit for each class model was assessed using the following fit criteria [42,43]: (1) the Bayesian information criterion (BIC), (2) the Akaike information criterion (AIC), (3) the likelihood/deviance ratio (G2) and its significance (significant results indicating poor model fit), (4) an adequate subsample size (greater than 10% of the total sample), and (5) the interpretability of classification results. Although BIC is the most accurate model fit measure because it considers the balance of parsimony and data fit [41], changes in both AIC and BIC were used to compare models.

## 3. Results

### 3.1. Sample Characteristics

The final dataset consisted of 632 valid cases after deleting incomplete cases (see Table 1). The sample comprised 51.7% females, and had an even age distribution (Range = 18–67 years, M = 31.29, SD = 9.06). The majority of respondents (81.8%) had a bachelor’s degree and their monthly family income followed a normal distribution, with 10.8% reporting a monthly income below CNY 6000, 21.0% stating that it is in the range of CNY 6001–10,000, 37.3% between CNY 10,001 and 20,000, 26.7% between CNY 20,001 and 60,000, and 4.1% over CNY 60,000.

### 3.2. WeRun Use and Exercise Behaviors

As indicated in Table 2, 72.8% of the sample population tended to check their WeRun step count rankings at least once a day and almost 80% often or always liked others’ step counts. Interestingly, only 50% of the respondents indicated that they were often or always liked by others. Meanwhile, only 38.4% of the sample population often or always donated step counts to charity, and 36.5% indicated that they sometimes, often, or always monetized their step counts. Moreover, 44.3% of the sample population reported performing moderate- to high-intensity physical activity at least three times per week; 55.4% exercised for longer than one hour each time, and 51.6% walked more than 7000 steps per day over the past week. To prepare the data for the latent class analysis, we assigned dummy codes for each item based on the scaled states that can best facilitate the process of differentiation [40,41,42] (see Table 2 for the assigned codes).

### 3.3. Latent Class Analysis

As can be seen from Table 3, several LCA models comprising 1–6 classes were estimated. The two-class model yielded the lowest BIC, but the three-class (ΔBIC = 9.165, *p* = 0.422) and the four-class (ΔBIC = 28.719, *p* = 0.052) models were equally parsimonious. However, the four-class model was superior to both the two-class (ΔAIC = 51.682, *p* < 0.001) and three-class (ΔAIC = 20.641, *p* < 0.05) models, for which the G2 statistics were significant. Thus, the four-class model provided the best fit for the data. A detailed examination of classification across the three models indicated that the two-class model categorized the sample into light/heavy users, while the four-class model further divided the heavy users into three interpretable subcategories with diverse behavioral patterns. As the four-class model provided more meaningful nuances, it was selected as the optimal model [41].

### 3.4. Latent Behavior Classes

After model selection, each latent class was labeled by its main WeRun use characteristics and exercise behaviors (see Figure 1). Table 4 presents the item-response probabilities for performing each behavior among the four classes.

Class 1, labeled as light users, included 171 participants (30.5%). This group were moderately likely to check step count rankings, but they had a lower likelihood of liking and receiving likes on step counts, donating steps to charity, and monetizing their step counts in WeRun. Moreover, their likelihood of walking more than 7000 steps per day, partaking in a moderate/high-intensity physical activity at least three times per week, and exercising for longer than an hour was, in general, low.

Class 2, labeled as reward-oriented users (27.5%), comprised moderate WeRun users who demonstrated strong interest in accumulating step counts. This group was more likely to check step count rankings, donate steps to charity, and monetize their step counts, but were less interested in interacting with other users, as evidenced by their lower likelihood of liking others and receiving likes from others. Their participation in physical activity was also step-based, whereby they had a higher likelihood of walking more than 7000 steps per day and exercising for longer than an hour per day, but were less likely to engage in moderate/high-intensity physical activity at least three times per week.

Class 3, labeled as lifestyle-oriented users (24.7%), included intensive WeRun users with high levels of physical activity. As illustrated in Figure 1, they had a higher likelihood of engaging in a full set of WeRun activities (i.e., checking, liking, receiving likes, monetizing, and donating) and their exercise behaviors were not limited to step-based exercises. In addition to walking more than 7000 steps per day, they were also more likely to exercise for longer than an hour per session and perform moderate- to high-intensity exercise three times per week.

Class 4, labeled as interaction-oriented users (17.3%), included moderate WeRun users that differed from reward-oriented users (who had a higher likelihood of donating and monetizing step counts, but a lower likelihood of interacting with WeRun friends) in that these individuals had a higher likelihood of liking each other, but they were less interested in donating steps to charity and monetizing step counts.

### 3.5. Covariate Analysis

In order to predict the class membership of WeRun users, we expanded our model to include the demographic variables and theoretically relevant variables as covariates. As can be seen from Table 5, gender, age, education, income, life satisfaction, autonomy, and the five platform-based motivations were all significant covariates of class membership.

Table 6 presents the covariate analysis results with light users as the reference group. It is evident that the participants with greater perceived autonomy had a higher likelihood of being lifestyle-oriented users (OR = 2.01, *p* < 0.05) than of being light users, whereas those that reported greater life satisfaction were more likely to be reward-oriented users (OR = 0.75, *p* < 0.05) than light users.

Overall, platform-based motivations were predictive of class membership, albeit with notable differences among patterns. First, participants with higher exercise motivations were more likely to be lifestyle-oriented users (OR = 1.62, *p* < 0.05) or interaction-oriented users (OR = 1.66, *p* < 0.01) than light users. Similarly, participants who reported higher social motivations had higher odds of being reward-oriented users (OR = 1.14, *p* < 0.05), lifestyle-oriented users (OR = 5.76, *p* < 0.001), or interaction-oriented users (OR = 3.10, *p* < 0.001) than of being light users, which suggests that these three classes are driven by social motivations. Between-group comparisons, as shown in Table 5, further indicate that lifestyle-oriented users reported the highest social motivations, followed by interaction-oriented users, reward-oriented users, and light users. Participants with higher financial motivations had higher odds of being reward-oriented users (OR = 1.17, *p* < 0.001) or lifestyle-oriented users (OR = 1.22, *p* < 0.01) than of being light users. Similarly, those with higher charity motivations had higher odds of being reward-oriented users (OR = 1.13, *p* < 0.01) or lifestyle-oriented users (OR = 2.70, *p* < 0.01), but lower odds of being interaction-oriented users (OR = –1.57, *p* < 0.01). Although comparison motivations were not predictive of class memberships, lifestyle-oriented users demonstrated significantly higher comparison motivations than the other three groups (see Table 5).

Analyses further revealed that females had a lower likelihood of being reward-oriented users (OR = −1.07, *p* < 0.05) or lifestyle-oriented users (OR = −1.75, *p* < 0.05) than of being light users. Older participants (OR = 0.07, *p* < 0.001) and those with higher incomes (OR = 0.58, *p* < 0.05) were more likely to be interaction-oriented users. Better educated individuals (OR = 1.86, *p* < 0.05) and those with higher incomes (OR = 2.06, *p* < 0.001) had a higher likelihood of being lifestyle-oriented users (OR = 1.86, *p* < 0.05).

## 4. Discussion

Our analyses revealed that, based on their app use characteristics and exercise behaviors, WeRun users can be classified as light, reward-oriented, lifestyle-oriented, or interaction-oriented users. While less than one-third of the users were categorized as light users (who use the app solely to check their step counts), moderate WeRun users can be segregated into reward-oriented (due to their preference for monetizing steps and donating steps to charity) and interaction-oriented users (due to their reliance on WeRun for social interactions, such as liking others’ step counts and/or receiving likes from others). Nonetheless, both reward-oriented and interaction-oriented users are more likely than light users to walk more than 7000 steps a day and exercise for longer than one hour. Finally, lifestyle-oriented users are not only the most physically active (i.e., they walk more than 7000 steps per day, exercise for longer than one hour, and engage in moderate/high-intensity exercise at least three times a week), but also use the widest range of app features. This type of user is more likely to have a higher level of education and higher income compared to light users. We interpret this high engagement with the fitness app and physical activity as a manifestation of a technology-facilitated healthy lifestyle.

The four fitness app user classes identified in our study are supported by extant evidence indicating the presence of substantial heterogeneity in app usage patterns, motivations and gratifications, and exercise behaviors [12,14,18,21,33,34]. However, in prior research, heterogeneity tended to be examined from a single perspective, such as demographics [12] or goal orientation [13]. Therefore, by incorporating a wide range of individual variables—including app use behaviors, exercise behaviors, motivations, and autonomy—the present study advances this line of research and provides a more nuanced understanding of fitness app usage and its association with individual characteristics. Moreover, based on the observed usage patterns, we argue that there might be a positive association between the number of app features users engage with and their physical activity level. Although this observation was not formally tested in the present study, Huang and Zhou [30] established that fitness apps that provide a full set of features guided by theories of health behavior change are more popular among Chinese users compared to apps that only focus on one or two behavior-change mechanisms. We extend this line of research by suggesting that users with different gratifications are motivated by different features (e.g., interaction-oriented users may be primarily driven by social motivations). Consequently, when designing fitness apps, it is important to account for the heterogeneity in user characteristics.

### 4.1. Practical Implications

The findings have significant practical implications for health promotion and app design practices. This study demonstrated the need for segmentation in understanding fitness app users. Prior practices regarding fitness app users only attend to the common characteristics of this group, such as weight concerns and health consciousness [4]. Based on the findings of this study, we suggest that app designers take the specific motivations and gratifications of each segment of app users into consideration and design appropriate features accordingly, in order to enhance the effectiveness of fitness apps [21]. Additionally, new strategies for promoting physical activity may be employed to further cater to the needs of a specific segment of users [33,34]. For example, the current reward-related features in WeRun only involve step counts, such as monetizing steps and donating steps to charity. New features that go beyond step-based activities may be introduced to provide more attractive incentives to motivate reward-oriented users to engage in more types of physical activity [21,27]. From the user’s perspective, we recommend that app users try to engage with as many features as possible [30]. The features of fitness apps are designed in the light of theories of health-behavior change, and have been demonstrated to be effective in motivating physical activity and forming healthy lifestyles. Our findings indicate that lifestyle-oriented users are characterized by both high engagement with a full set of app features and high engagement with physical activity. Therefore, to increase the efficacy of using fitness apps, users should achieve high involvement using the whole range of app features.

### 4.2. Limitations

Although the present study demonstrates some important results, there are some limitations when interpreting the study findings. The first of these concerns its limited scope (targeting only users of WeRun, an app embedded in China’s most popular mobile social networking app—WeChat). Thus, further studies are needed to validate the findings in a wider range of fitness apps and other cultural contexts. Moreover, as WeRun shares WeChat’s user contacts, it benefits from a prebuilt user community for social functions. As such, it would be interesting to examine how the usage patterns of apps that are part of a wider app ecology differ from those of stand-alone apps. A further limitation arises from the sample selection being conducted via an online platform, rendering it non-representative of all Chinese WeRun users, or other fitness app users. Third, this study mainly used self-reported data and one two-item scale (exercise motivation), demonstrated low reliability. Thus, more robust measures, such as multi-item scales and behavior tracking data, should be used in future investigations to gain a more accurate picture of app usage.

## 5. Conclusions

Fitness apps have become a common mobile technological aid for popularizing physical activity and promoting a healthy lifestyle. Motivated by the growing number of fitness app users, we set out to categorize them based on app use and exercise behaviors. This resulted in four classes: light users (30.5%), reward-oriented users (27.5%), lifestyle-oriented users (24.7%), and interaction-oriented users (17.3%). We further revealed that platform-based motivations (i.e., exercise, social, financial, charity, and comparison) are significant predictors of user classification, echoing the uses and gratification theory in that the more gratification people receive from the app, the more engaged they are with the app. We also found that demographics and other individual characteristics are relevant factors for class membership.

## Figures and Tables

**Figure 1 ijerph-19-10679-f001:**
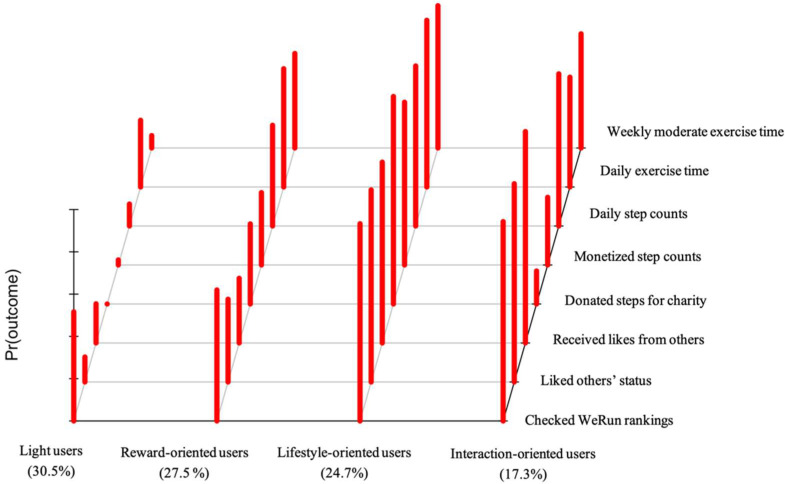
Conditional item response probabilities for the eight behavioral items in both classes.

**Table 1 ijerph-19-10679-t001:** Sample characteristics of the present study (N = 632).

Variables	N (Percentage)
** *Age* ** *(Mean, SD)*	31.29 (9.06)
** *Gender* **	
Males	305 (48.3)
Females	327 (51.7)
** *Education* **	
Some high School or below	2 (0.3)
High school graduate	40 (6.3)
College graduate	517 (81.8)
Masters graduate or above	73 (11.6)
** *Family* ** * **income** *	
Lower than CNY 6,000	68 (10.8)
CNY 6001 ~ 10,000	133 (21.0)
CNY 10,001 ~ 20,000	236 (37.3)
CNY 20,001 ~ 60,000	169 (26.7)
Over CNY 60,000	29 (4.1)

1 CNY ≈ 0.13 USD.

**Table 2 ijerph-19-10679-t002:** Indicators for latent class analysis (N = 632).

	Code	Label	%
Check step count rankings	1	Never, occasionally, 2–3 times per week	27.2%
	2	At least one time per day	72.8%
Like others’ step counts	1	Never, occasionally, sometimes	20.7%
	2	Often, always	79.3%
Receive likes from others	1	Never, occasionally, sometimes	46.8%
	2	Often, always	53.2%
Donate steps to charity	1	Never, occasionally, sometimes	61.6%
	2	Often, always	38.4%
Monetize their step counts	1	Never, occasionally	63.5%
	2	Sometimes, often, always	36.5%
Perform moderate/high-intensity exercises	1	Less than three times per week	55.7%
	2	At least three times per week	44.3%
Walk more than 7000 steps per day	1	No	48.4%
	2	Yes	51.6%
Exercise for longer than an hour	1	No	44.6%
	2	Yes	55.4%

**Table 3 ijerph-19-10679-t003:** Model fit information for the comparison of latent class models.

Class	LL	AIC	BIC	G2	df	*p* Value	Parameters	Class Memberships
1	–3383.311	6782.623	6818.214	714.460	247	<0.00001	8	100%
2	–3241.111	6516.223	6592.147	335.68	238	0.00003	17	1: 52.8% 2: 47.2%
3	–3216.596	6485.192	6601.312	286.65	229	0.005758	26	1: 47.9% 2: 32.2% 3: 19.8%
4	–3197.275	6464.551	6620.866	248.01	220	0.094425	35	1: 30.5% 2: 27.5% 3: 24.7% 4: 17.3%
5	–3183.685	6455.369	6651.880	220.83	211	0.307265	44	1: 21.5% 2: 23.8% 3: 21.3% 4: 15.9% 5: 17.4%
6	–3174.213	6454.426	6691.131	201.887	202	0.489148	53	1: 6.50% 2: 22.1% 3: 21.4% 4: 27.0% 5: 18.3% 6: 4.80%

**Table 4 ijerph-19-10679-t004:** Item-response probabilities for the four-class model, given latent class membership.

	Light Users	Reward-Oriented Users	Lifestyle-Oriented Users	Interaction-Oriented Users
Percentage	193 (30.5%)	174 (27.5%)	156 (24.7%)	109 (17.3%)
Check step count rankings	**0.54**	**0.63**	**0.96**	**0.89**
Like others’ step counts	0.23	0.40	**0.99**	**0.79**
Receive likes from others	0.21	0.42	**0.90**	**0.79**
Donate steps to charity	0.11	**0.58**	**0.79**	0.00
Monetize their step counts	0.04	**0.59**	**0.65**	0.17
Do moderate/high-intensity exercises three times/week	0.21	0.47	**0.63**	**0.56**
Walk more than 7000 steps per day	0.21	**0.52**	**0.78**	**0.70**
Exercise for longer than an hour	0.40	**0.61**	**0.71**	**0.52**

Item-response probabilities > 0.50 in bold to facilitate interpretation.

**Table 5 ijerph-19-10679-t005:** Covariate analysis with light users as the referent class.

	Rewards-Oriented Users			Lifestyle-Oriented Users			Interaction-Oriented Users		
	*Coefficient*	*SE*	*t*	*Coefficient*	*SE*	*t*	*Coefficient*	*SE*	*t*
Female	−1.07 *	0.44	−2.41	−1.75 *	0.73	−2.41	−0.40	0.51	−0.80
Age	0.01	0.03	0.51	0.04	0.04	0.95	0.07 *	0.03	2.47
Higher education	0.65	0.54	1.21	1.86 *	0.92	2.02	0.18	0.56	0.31
Higher income	0.19	0.22	0.86	2.06 ***	0.49	4.19	0.58 *	0.29	2.00
Life satisfaction	0.75 *	0.43	1.29	0.54	0.42	1.28	0.49	0.35	1.40
Autonomy	0.70	0.53	1.32	2.01 *	0.86	2.32	−0.43	0.56	−0.78
Exercise motives	0.73	0.38	1.94	1.62 *	0.76	2.13	1.66 **	0.50	3.31
Social motives	1.14 *	0.48	2.39	5.76 ***	1.30	4.43	3.10 ***	0.63	4.93
Financial motives	1.17 ***	0.29	4.02	1.22 **	0.42	2.93	−0.01	0.34	−0.02
Charitable motives	1.13 **	0.37	3.01	2.70 **	0.80	3.37	−1.57 **	0.52	−3.02
Comparison motives	0.21	0.32	0.64	0.63	0.56	1.13	0.52	0.39	1.35

* *p* > 0.05; ** *p* < 0.01; *** *p* < 0.001.

**Table 6 ijerph-19-10679-t006:** Fitness users’ characteristics across four classes.

	Class 1 Light Users	Class 2 Reward-Oriented Users	Class 3 Lifestyle-Oriented Users	Class 4 Interaction-Oriented Users	*p*-Value
Count (%), total n = 632	193 (30.5)	174 (27.5)	156 (24.7)	109 (17.3)	
** *Sex* **					
Male	72 (23.6)	102 (33.4)	75 (24.6)	56 (18.4)	0.000
Female	121 (37.0)	72 (22.0)	81 (24.8)	53 (16.2)	
** *Education* **					
Some high School or below	2 (100)	0 (0.0)	0 (0.0)	0 (0.0)	0.000
High school graduate	12 (30.0)	13 (32.5)	6 (15.0)	9 (22.5)	
College graduate	164 (31.7)	148 (28.6)	111 (21.5)	94 (18.2)	
Masters graduate or above	15 (20.5)	13 (17.8)	39 (53.4)	6 (8.2)	
** *Family income* **					0.000
Lower than CNY 6000	39 (57.4)	22 (32.4)	1 (1.5)	6 (8.8)	
CNY 6001 ~ 10,000	51 (38.3)	41 (30.8)	23 (17.3)	18 (13.5)	
CNY 10,001 ~ 20,000	63 (26.7)	80 (33.9)	49 (20.8)	44 (18.6)	
CNY 20,001 ~ 60,000	34 (20.1)	27 (16.0)	71 (42.0)	37 (21.9)	
Over CNY 60,000	6 (23.1)	4 (15.4)	12 (46.2)	4 (15.4)	
*Mean (SD)*					
** *Age* **	27.95 ^a^ (9.38)	29.78 ^a^ (7.72)	33.95 ^b^ (7.27)	35.83 ^b^ (9.87)	0.000
** *Life satisfaction* **	2.69 ^a^ (0.81)	3.35 ^b^ (0.77)	3.89 ^c^ (0.94)	3.28 ^b^ (0.89)	0.000
** *Autonomy* **	3.20 ^a^ (0.64)	3.87 ^b^ (0.48)	4.31 ^c^ (0.31)	3.92 ^b^ (0.50)	0.000
** *Exercise motives* **	3.36 ^a^ (0.77)	3.91 ^b^ (0.58)	4.40 ^d^ (0.44)	4.19 ^c^ (0.49)	0.000
** *Social motives* **	3.00 ^a^ (0.63)	3.66 ^b^ (0.45)	4.29 ^d^ (0.31)	3.83 ^c^ (0.49)	0.000
** *Financial motives* **	2.85 ^a^ (0.81)	3.50 ^b^ (0.77)	3.70 ^b^ (0.94)	2.72 ^a^ (0.89)	0.000
** *Charitable motives* **	3.28 ^b^ (0.71)	3.87 ^c^ (0.58)	4.36 ^d^ (0.44)	3.00 ^a^ (0.76)	0.000
** *Comparison motives* **	3.82 ^a^ (0.78)	4.40 ^b^ (0.71)	5.07 ^c^ (0.72)	4.45 ^b^ (0.74)	0.000

Different subscripts within rows are significant at *p* < 0.05 based on Bonferroni-adjusted pairwise comparisons.

## Data Availability

Not applicable.

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
