# Peer review of "Uncovering the Heterogeneity in Fitness App Use: A Latent Class Analysis of Chinese Users"

_ijerph, 2022, doi:10.3390/ijerph191710679_

Round 1
Reviewer 1 Report
This study focuses on the heterogeneity in fitness app usage patterns using a latent class analysis.
1. In the second paragraph of the introduction, the authors explained, "relatively little is known about app usage patterns and their potential link to specific types of physical activity." However, the following sentence contradicts the previous sentence. Please re-organize the sentences to avoid any confusion
2. The following sentence is also a bit confusing for potential readers.
"Therefore, to achieve a more nuanced understanding of how fitness apps may motivate physical activity, it is necessary to differentiate fitness app usage patterns and correlate them to individual traits and motivations."
I don't think two things have closely related to each other – 1) fitness app usage patterns and 2) the understanding of how fitness apps motivate physical activity.
For the second part, the authors should directly correlate fitness app use (or motivation) and physical activity.
Thus, the authors should re-consider the purpose of the current research and provide a better justification of why this study is needed.
3. The authors explained that sample recruitment focused on maximizing sample representativeness by covering different age and gender groups. Please explain further what kinds of sampling methods were used.
4. In this study, five items of fitness app use behaviors have an essential role in creating a werun user profile. Please explain how to identify five behavioral items in much detail.
5. Pearson's correlation for exercise motivations was relatively low (0.39). Please justify.
6. For the data analysis part, the authors explained that werun use and exercise behaviors are shown in Table 1. However, it might be table 2. Please check and revise.
7. For table 2, the authors recode variables by collapsing five scales into two. For three variables (e.g., like others' step counts, receive likes from others, and donate steps for charity), 'never' 'occasionally,' and 'sometimes' were coded '1' and 'often' and 'always' were coded for '2'. However, for another variable (monetize their step counts), 'never' and 'occasionally' were coded for '1' and 'sometimes,' 'often' and 'always' were coded for '2'. Please explain why two recoding strategies were different and justify.
8. Figure 1 does not clearly show the differences in behaviors between groups. Please consider changing figure 1.
9. Tables 5 and 6 indeed show the same results differently. Please justify why two methods were conducted and explain the results again.
Reviewer 2 Report
Nowadays, fitness applications are growing in popularity and therefore it is important to recognize the determinants of their effectiveness, so that their usefulness increases and that a healthy lifestyle is promoted. Such issues were discussed in the reviewed article. In order to recognize how fitness applications can motivate to physical activity, it is necessary to analyze the patterns of using this type of application, which are the correlations with the individual characteristics of users and their motivations. The authors have included this problem in three research questions, as well as in the detailed objective, which concerns the practical implications of designing fitness applications.
However, the introduction of a precisely defined main research goal is missing. The aim of the study was only defined in chapter 2.1. but it is also not stated explicitly that this is the main purpose of the study. Therefore, I suggest that the main research goal should be precisely defined in the introduction. There is also a question why no research hypothesis has been formulated. Research methods and tools have been described precisely and correctly.
I suggest that the comment to table 1 should be placed below it, and it is in the next chapter 3.1.
Table 2 should be in my opinion in the chapter "WeRun Use and Exercise Behaviors".
No comment in chapter 3.5 to figure 1.
It would be advisable to expand the literature used in the discussion of the results.
In conclusion, I propose to indicate specific numbers resulting from the conducted research regarding the main goal and research questions.
I share the opinion that it is advisable to continue research using other applications (limited to WeRun in the article) and demographically diverse trials to increase public interest and promote a healthy lifestyle.
